## RESEARCH CULTURE

# Surveying the experience of postdocs in the United States before and during the COVID-19 pandemic

**Abstract** In the interest of advocating for the postdoctoral community in the United States (US), we compared the results of surveys of postdocs carried out in 2019 and in late 2020. We found that respondents' mental health and wellness were significantly impacted by the pandemic irrespective of their gender, race, citizenship, or other identities. Career trajectories and progression were also affected, as respondents reported being less confident about achieving career goals, and having more negative perceptions of the job market compared to before the pandemic. Postdocs working in the US on temporary visas reported experiencing increased stress levels due to changes in immigration policy. Access to institutional Postdoctoral Offices or Associations positively impacted well-being and helped mitigate some of the personal and professional stresses caused by the pandemic.

**ANDRÉANNE MORIN**[*][†], **BRITNEY A HELLING**[†], **SEETHA KRISHNAN**[†], **LAURIE E RISNER, NYKIA D WALKER AND NANCY B SCHWARTZ**[*]

**\*For correspondence:**
amorin@uchicago.edu (AM);
n-schwartz@uchicago.edu (NBS)

[†]These authors contributed equally to this work

**Competing interest:** The authors declare that no competing interests exist.

## Introduction

Postdocs have long been referred to as the invisible component of the University (*National Research Council, 1969*). Although they comprise a significant portion of the scientific workforce, postdocs often lack job security, receive lower pay in comparison to non-academic peers in government or industry, and frequently lack employee-type benefits such as paid family leave (*Shen, 2015*; *Alund et al., 2020*).

The COVID-19 pandemic has only made things worse for postdocs due to lab closures, rotating (work) schedules, and hiring and salary freezes. Things were especially difficult for post-docs with families, as school and daycare closures significantly disrupted research continuity (*Park, 2020*; *National Academies of Sciences, Engineering, and Medicine, 2021*). For example, early in the pandemic it was reported that nearly two-thirds of postdocs believed that their long-term career prospects were negatively affected by the COVID-19 pandemic (*Woolston, 2020a*; *Woolston, 2020b*). Roughly eight out of ten postdocs surveyed also reported that the

pandemic had hampered their ability to conduct experiments and collect data, and more than half had difficulty communicating with supervisors and colleagues. Furthermore, numerous universities retracted or deferred new faculty job offers, leaving postdocs (who are the source of future academics) to either consider different career paths or extend their current positions.

The impact of the pandemic on postdocs is not unlike the severe and far-reaching effects observed throughout the biomedical workforce, such as significant job losses, educational disparities, and elevated mental health issues (*Levine et al., 2021*; *Jagsi et al., 2021*; *Gao et al., 2021*; *Doyle et al., 2021*; *Korbel and Stegle, 2020*; *Carr et al., 2021*; *Yan, 2020*; *Servick et al., 2020*; *Andrade et al., 2022*; *Myers et al., 2020*). Although the financial impact of COVID-19 on scientific productivity has not yet been fully realized, the National Institutes of Health (NIH) estimates a $16 billion loss because of delayed research, and the Bureau of Labor Statistics reported the largest decline in college and university employment since the

1950s (*Baumann, 2021*; *Bauman, 2021*; *Bauman, 2020*).

We have long been interested in the postdoctoral experience in the United States (US) with respect to career choices, mentorship, grants and gender disparities. In 2018, we published the first comprehensive survey of postdocs since 2005 (*Davis, 2005*), assessing the satisfaction and career plans of over 7,500 postdocs from 351 institutions across the US (*McConnell et al., 2018*). To continue tracking these aspects of the postdoc experience, we conducted a second survey of over 6,000 postdocs in 2019 (June to December). As the effects of the COVID-19 pandemic began to be felt widely, a follow-up survey was conducted in the Fall of 2020 (between October 1 and November 3) on a subset (n=1,942) of the 2019 survey respondents to assess the impact of the pandemic on this postdoc trainee population.

Here, we present a comparison of survey data collected before and during the COVID-19 pandemic. We investigated how the pandemic impacted postdocs' mental health and wellness, career trajectories and progression, and their confidence in achieving their career goals. The survey also explored specific challenges faced by international postdocs during the pandemic when the government enacted new policies restricting international travel, immigration, and visa access. Additionally, we enquired how access to Postdoctoral Offices (PDOs), which are run by professionals and funded by institutions, and Postdoctoral Associations (PDAs), which are largely managed by postdocs themselves, impacted the respondents' overall well-being during the pandemic. These institutional services represent the postdoc community and support their training and professional development, but their role in the pandemic had not been previously explored.

## Results

### Comparing demographic data from the pre-pandemic and pandemic survey

In 2019, 6,292 respondents participated in our national postdoc survey, of which 5,929 identified as postdocs working in the US (we only analyzed responses from US-based postdocs). In October of 2020, 1,942 of the 6,292 respondents who participated in the 2019 survey, completed a follow-up survey assessing the effects of the COVID-19 pandemic. Of these, 1,722 (89%) were still in a postdoctoral position at a US institution.

From here on, we refer to the 2019 survey as the pre-pandemic survey and the 2020 survey as the pandemic survey. Furthermore, in our analyses of current postdocs, we removed the 11% of respondents in the pandemic survey who were no longer in postdoctoral positions; however, we analyzed their career outcomes separately in Figure 7.

As shown in *Figure 1*, the demographics of the respondents to the pandemic survey largely mirrored those of the pre-pandemic survey. The number of respondents in the pandemic and pre-pandemic survey by US state is shown in *Figure 1—figure supplement 1A-B*. There were slightly more responses from individuals who identified as female (61% vs 58%) and non-binary/third gender (0.9% vs 0.4%), and fewer self-identified males (38% vs 42%) in the pandemic survey compared to the pre-pandemic survey (*Figure 1A*).

Race and ethnicity varied between the pre-pandemic and pandemic survey respondents, with a 4% increase in the proportion of respondents who identify as white, and a corresponding 4% decrease in the respondents who identify as Asian; however, no differences were observed between the proportion of respondents from underrepresented minority backgrounds (URMs; 13% in both surveys; *Figure 1B*; *Figure 1—figure supplement 1C*). We combined individuals into these three main categories (white, Asian and URM) due to the small number of respondents in some racial and ethnic groups (see Methods for a full description and *Supplementary file 3* for a more granular breakdown). The number of responses from other identity groups (i.e., disability, LGBTQ, and veterans) also did not vary between the two surveys (*Figure 1—figure supplement 1D*).

The number of respondents who were US citizens or permanent residents (referred to as US citizens/PR throughout this manuscript) increased from 53% (pre-pandemic survey) to 57% (pandemic survey). There was a corresponding decrease in international respondents (47% pre-pandemic vs. 43% pandemic) who were working in the US on temporary visas (J1, H1B, TN, F1, F1-OTP, E3 visas; *Figure 1C*). Given that we conducted the pandemic survey within a subpopulation of those in the pre-pandemic survey at a later date, the age of the pandemic respondents was slightly higher than the pre-pandemic respondents, and as expected, they were more advanced in their postdoc (*Figure 1D–E*).

There was a significant decrease in respondents in the field of medicine (13% pre-pandemic

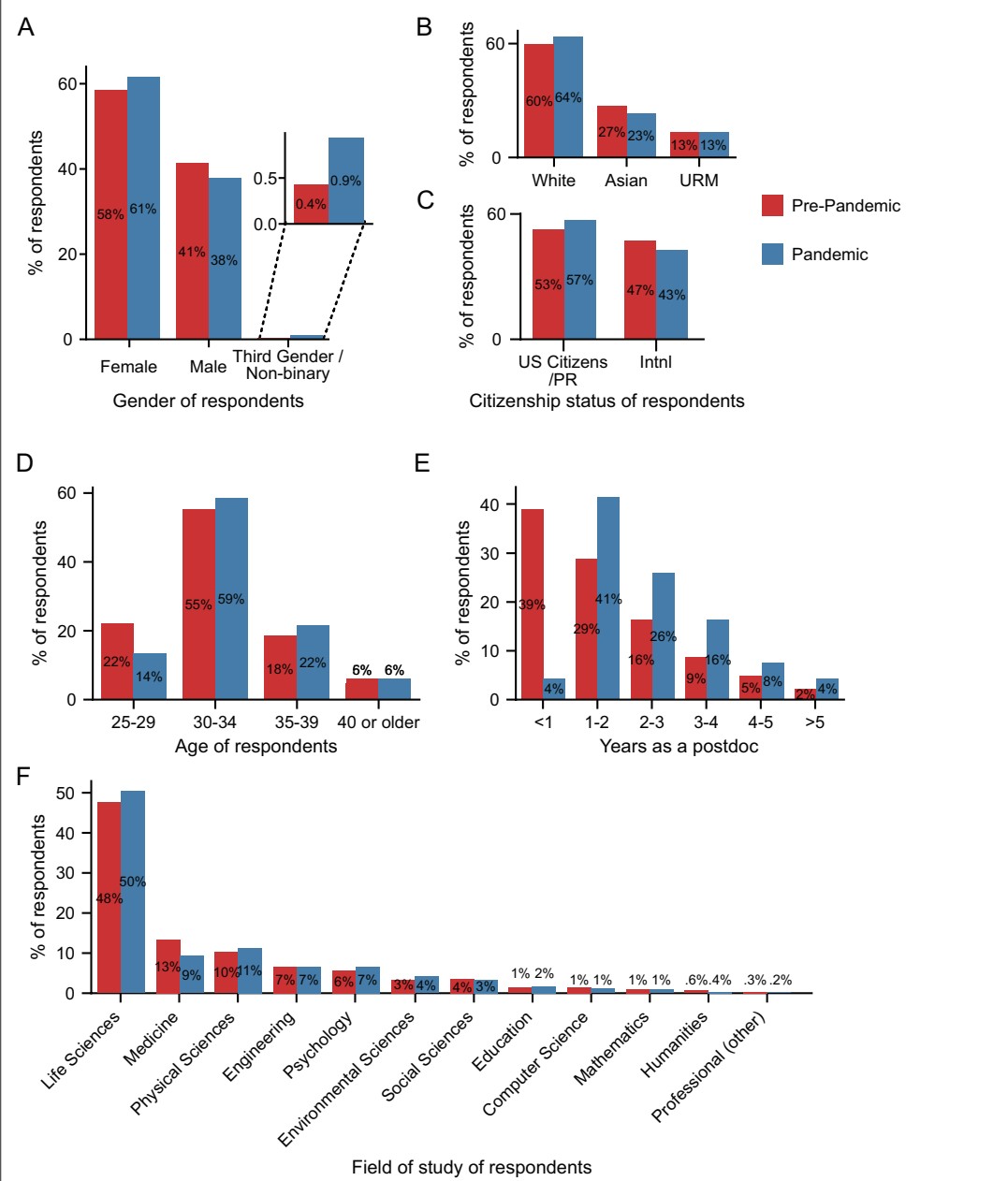

**Figure 1.** Pre-pandemic and pandemic survey demographics. (**A**) More self-identified female and third gender/ non-binary and fewer self-identified male respondents completed the pandemic survey (n=1,698) compared to the pre-pandemic survey (Chi-squared test, P=0.0023, $\chi 2$=12.2; n=5,805). (**B**) The majority of respondents were white in both the pre-pandemic (n=5,649) and pandemic surveys (n=1,673), with an increase in white and a decrease in Asian respondents in the pandemic survey compared to the pre-pandemic survey (Chi-squared test, P=0.0024, $\chi 2$=12.1). (**C**) The proportion of US citizens/PR respondents increased (Chi-squared test, P=0.0015, $\chi 2$=10.1; n pre-pandemic=5,813; n pandemic = 1,702). (**D–E**) As expected, the age of respondents (**D**) and the years of postdoc experience (**E**) both increased as the pandemic survey was conducty with a subset of the pre-pandemic respondents almost one year after the initial survey. (**F**) The majority of respondents were in the life sciences with a statistically significant decrease in responses from those in the field of medicine in the pandemic survey (n=1,712) compared to the pre-pandemic survey (Chi-squared test, P=0.0012, $\chi 2$=32.47; n=5,922). PR: Permanent Resident. Additional demographic information from the two surveys is shown in *Figure 1—figure supplement 1*.

The online version of this article includes the following figure supplement(s) for figure 1:

**Figure supplement 1.** Comparison of demographics between pandemic and pre-pandemic surveys.

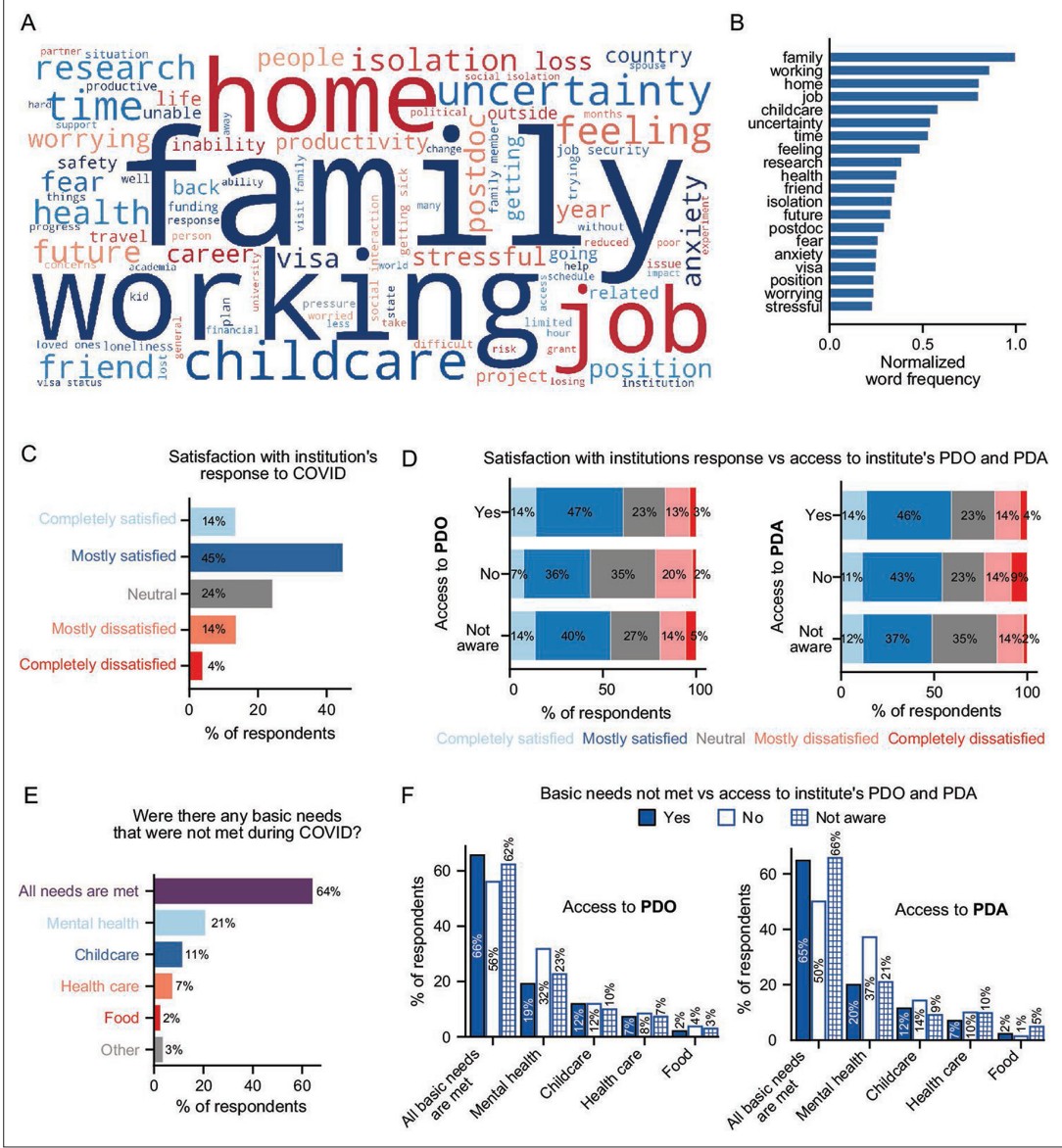

**Figure 2.** Impact of the pandemic on postdocs and the effect of institutional support. (**A-B**) Word cloud of postdocs' main stressors during the COVID-19 pandemic (**A**) and distribution of the most frequently used words (**B**). (**C**). Satisfaction with the institution's response to COVID-19 (n=1,718). (**D**) Satisfaction with the institution's response to COVID-19 was higher in postdocs that had access to a PDO compared to the ones that did not have a PDO at their institution (multivariate ordinal logistic regression P=0.0021, Odds Ratio (OR)=1.81 [95% Confidence Interval (CI); 1.24–2.65]; n=1,613) or those unaware whether their institution had a PDO (multivariate ordinal logistic regression P=0.044, OR=1.24 [95% CI; 1.01–1.53]; n=1,700). No significant differences were observed by access to PDA. (**E**) Basic needs that were not met during the pandemic (n=1,676). See *Figure 2—figure supplement 1* for breakdown by race/ethnicity and identity groups. (**F**) Having access to a PDO or a PDA significantly impacted having mental health needs met (multivariate logistic regression, No PDO P=0.02, OR=0.57 [95% CI; 0.35–0.91]; No PDA P=0.026, OR=0.52 [95% CI; 0.29–0.93]; n=1,614).

The online version of this article includes the following figure supplement(s) for figure 2:

**Figure supplement 1.** Basic needs not met broken down by demographic data.

and 9% pandemic), while there was no significant change in the representation of any other field (*Figure 1F*). Respondents were from various disciplines, mostly within the life sciences (48%), followed by medicine, physical sciences, engineering, psychology, environmental sciences, and social sciences, among other research areas. Lastly, there was a significant increase in access to

a PDO (65.6% pre-pandemic vs 70% pandemic), which was mainly due to increased awareness, but no differences in access to a PDA (*Figure 1—figure supplement 1F-G*).

### The impact of COVID-19 on postdoc well-being

To directly assess the effects of COVID-19 on postdocs, we queried three general areas: stressors during the pandemic, institutional response to the pandemic, and ability to meet basic needs. In an open-ended question inquiring about main stressors during the pandemic, postdocs indicated that their main stressors were a combination of work, family, and emotional burdens (such as uncertainty, isolation, fear, and anxiety) as shown by the word cloud analysis of the responses (*Figure 2A–B*). Additional responses to the open-ended question are included in *Table 1* along with responses to another question querying how the pandemic impacted research productivity. 2,768 comments were collected in total, which further demonstrates the impact of the pandemic on the postdoctoral population.

Individual responses showed how postdocs experienced different types of burdens. Parents and caregivers faced the burden of "being a full time [*sic*] postdoc and staying home with two kids" or caring for a loved one who was/is struggling with COVID-19. As one postdoc indicated, "my girlfriend has been recovering from COVID-19 since March. It's a grueling process to watch and support." A large number of postdocs also indicated that work progress was more difficult due to "getting research done within limited shifts and hours" and an overall fear of "loss of productivity". Many international postdocs were concerned about their visas and one respondent even indicated that the international office at their institution told them "...you will lose your job if you leave the country for any reason and are not a resident."

Laboratory shutdowns also had an adverse impact on postdocs' relationships with their Principal Investigator (PI) and coworkers. When asked if respondents were able to maintain regular contact with their PI and coworkers, half of the respondents (50%) reported they had, but not as much as before the pandemic and 1% reported no contact (49% reported maintaining as much contact as before the pandemic). In open-ended responses, postdocs indicated facing high demands from PIs and unrealistic expectations to be productive during the pandemic (examples in *Table 1*). Some felt that work from home was

expected to be "business as usual" and there was immense pressure to "work round the clock", "work long hours and continuously produce results" and "produce data when no lab activities are allowed". One respondent indicated inability to utilize institutional support services due to over-work: "the PI puts a large amount of pressure and therefore there is really no time to make use of any of the resources".

Conversely, supportive PIs were lauded for their role in lessening stress. Respondents mentioned: "I did not have a lot of stress factors. I was lucky to have a supportive PI that understood how stressful a time this can be and set a pretty low expectation bar"; "working from home during shutdown with a 5 yo kid was impossible, really stressful and I am happy my PI was understanding and let me work half time."

### Institutional responses to COVID-19 and the impact of PDAs and PDOs

Next, we examined the institutional response to COVID-19, which ranged from completely satisfied to completely unsatisfied. Most postdocs indicated that they were completely or mostly satisfied with their institution's response to COVID-19 (59%) (*Figure 2C*). In particular, postdocs with access to a PDO were significantly more satisfied than those who did not (*Figure 2D*). On the other hand, there were no differences in satisfaction with their institution's response between those with or without access to a PDA (*Figure 2D*), or with respect to gender, citizenship status, race and ethnicity, or identity (multivariate ordinal logistic regression, $P>0.05$, data not shown). Notably, there was also a non-negligible portion (4%) of postdocs who indicated they were completely unsatisfied with their institution's response to COVID-19, with one respondent commenting, "... my institution did almost NOTHING to ensure that faculty and staff can be safely back at work".

### Meeting basic needs during the pandemic

Although the majority of postdocs indicated that all of their basic needs were met during the pandemic (64%), a substantial portion (36%) indicated that their needs concerning mental health (21%), childcare (11%), healthcare (7%) and/or food (2%) were unmet (*Figure 2E*). Additionally, 3% of postdocs mentioned other unmet needs in their responses, including the inability to pay bills or exercise, and loss of access to transportation, work safety, human connections, or loss of salary,

**Table 1.** Response to open questions.

Selected responses to the questions "Why or how has your research been disrupted (or not disrupted) due to the pandemic?" and "What were your main stress factors during the pandemic?" in the pandemic survey.

**Mental Health**

Uncertainty in my health, uncertainty in my partner's health, anxiety about leaving home, anxiety about how this will affect my future, depression and grievance of lost sense of "normal", lack of social interaction with others, can't visit family for forseeable [sic] future, lack of sufficient space to work from home productively, stress of fighting institutionalized racism, anxiety over changing career prospects.

Loss of morale, loss of collegial atmosphere, perception that the world is going to end, chronic anxiety about the US political situation, minority stress, worry about the health of family members, realization that working alone is terrible for my mental health, realization that nobody reads academic articles and nobody respects the professoriate, realization that the general public does not believe in science or truth.

My mental health has suffered as a consequence of being alone all the time making research more difficult….

…the extra stressors associated with the pandemic have significantly affected my mental health and ability to work effectively.

…The pandemic has also taken a huge toll on my mental health which has disrupted my focus and ability to get research done.

**Immigration/ International postdocs**

The government released multiple rules controlling the H1-B visa of foreign workers, which make it harder for us foreigners in the job market.

1. Family getting sick and dying back home in India due to COVID-19, 2. Immigration restrictions by the government, 3. Slow pace of immigration application procedures by USCIS and US Embassies…

As I a [sic] here in the US alone. My stress came from being worried about my family back in my country. and in experiencing this pandemic nearly all alone.

Having the pandemic eat into the limited amount of time I have as a postdoc here. Also being unable to travel - due to the travel ban, I cannot return home to see family (e.g. for Christmas) because I wouldn't be able to get back into the US.

I was stuck in Europe for 6 months due to immigration issues (expired visa and closed embassies) and therefore was not able to do any lab work.

**Relationship with PI**

I have been working from home, which has led to a drop in productivity. However, my PI expects me to be more productive due to "a lack of distractions." This disparity is making progress difficult….

Personally, my research has been disrupted by the constant pressure by my PI and my Institution to continue to work in lab during a pandemic. I don't feel safe working around so many people, and my complaint has been ignored by my PI and the Institution. This has caused me a lot of stress and anxiety.

… My supervisors also fell off of the map and we had almost zero contact throughout the lockdown (March - June) until we could return to the lab. Then after, the communication is still minimal and it's unclear what the status of publications are.

My PI became very micromanaging, in stark contrast to her hands-off style previously. They put a lot of pressure on me to publish and be productive during the pandemic.

Unrealistic expectations of the PI who ignored/ignores the fact that there is a pandemic and that the pandemic has an impact on research progress. First, the lab was shut down and then reopened with 25% capacity at a time.

**Career/job perspectives**

Uncertainty/Instability in the job market as I try to find a job… Poor postdoc pay relative to the job market for my degree & experience level.

… Feeling like industry/private sector is not going to be any easier to find employment in than academia with such high unemployment rates …

That my project is getting behind and I will not be able to apply for grants within the window of "early career"/trainee grants.

Lack of career perspective and being unable to do my research during the final years of my postdoc.

**Research Productivity**

I was expected to continue producing lab work while the labs were closed down! My PI encouraged me to break quarantine rules and continue work.

Lack of research output leading to fears of my career being over.

The feeling of guilt has been overwhelming. I feel like I should be doing more, but I really can't because I don't have the resources needed (e.g. mice) to do my research.

… trying to find new ways of ensuring/displaying productivity. I couldn't produce experimental results so how to represent the work that I've actually been getting done during this time. and [sic] then upon start-up, are they actually concerned and keeping student/worker safety as their primary goal.

**Family/Childcare**

Lockdown forced to ramp-down research to the bare minimum. Childcare restrictions have also impacted the amount of time that I can spend in the lab. Taking care of a toddler at home does not favor literature research.

An inability to balance work with childcare. My wife worked full or nearly full-time throughout the pandemic, and as a result, the bulk of childcare fell on me because I had a more flexible schedule and understanding PI. I constantly felt pressure and stress to accomplish research goals but consistently was unable to achieve anything because my children's welfare was top priority.

Lack of childcare for my school-age child. Non-COVID health concerns for my household members and paying for co-insurance and copays with the terrible insurance of my institution. My husband is unemployed and can find safe work and we are financially struggling.

*Table 1 continued*

**Family/Childcare**

Loss of productivity due to loss of childcare, feeling like I am slipping behind my colleagues without children. Lots of stress and pressure around keeping up with tasks. Unable to start any new, exciting projects that would help my career due to childcare loss.

Trying to work from home while caring for my children; it's like normal working mom guilt, but on steroids. Also, the university permanently closed the childcare center on campus (one of the best centers in the area) where our children went, so the uncertainty of being able to find quality childcare once centers reopened was exceptionally stressful.

retirement benefits, or annual raise. Furthermore, although the majority of postdocs indicated that all of their basic needs were met, their comments suggested that the pandemic had made meeting those needs more difficult: "My husband lost his job, and while we are not in danger of basic needs not being met it does change some things and adds additional stress".

Postdocs who had access to a PDA or PDO (although not statistically significant for the latter) were more likely to have their basic needs met (65% PDA; 66% PDO) than those who did not (50% PDA; 56% PDO; *Figure 2F*). Furthermore, postdocs with access to a PDO or a PDA were less likely to have their mental health needs unmet (PDO: 32% (no) vs. 19% (yes); PDA: 37% (no) vs. 20% (yes)); no differences were observed between those not aware and aware of a PDA or PDO at their institution (*Figure 2F*). Postdocs who identified as Asian – the majority of whom were international (76%) – were more likely than white postdocs to report unmet needs with respect to health care (12% vs 6%) or food (6% vs 1%) (*Figure 2—figure supplement 1A*). Postdocs who identified as LGBTQ reported an increase in food (5% vs 2%) and mental health (30% vs 20%) needs unmet compared to postdocs who did not identify as LGBTQ. (*Figure 2—figure supplement 1B*). The latter was also observed in postdocs with disabilities compared to postdocs without disabilities (40% vs 20%; *Figure 2—figure supplement 1C*). No differences were observed according to URM status (*Figure 2—figure supplement 1A*) or gender (multivariate logistic regression, $P>0.05$, data not shown).

Postdoc parents were particularly affected by pandemic-related shutdowns and reported difficulties meeting childcare needs. While we did not directly inquire whether respondents in the pandemic survey had children (in the pre-pandemic survey, 20% of postdocs answered that they had children), 10% of respondents mentioned in comments that ensuring their children had proper care was a major stressor and led to severe work disruptions. Additionally, 68% of these comments were from female respondents and 32% from males, suggesting a greater

burden of childcare for female postdocs. Overall, childcare was the 5th most frequently mentioned stressor (*Figure 2A–B*). Parents mentioned "I have lost childcare for my baby and it has had a significant impact on my ability to write, complete research goals, and apply for grants"; "It was difficult to do any writing- or reading-based work because the day cares were closed, and my partner and I had to divide the day into childcare/work time"; "Loss of productivity due to loss of childcare, feeling like I am slipping behind my colleagues without children". Some reported feeling burnt out from putting in long hours and mentioned lack of support from their peers and their university; "Lack of childcare and intense pressure from PI to continue long hours at home"; "Loss of childcare and co-workers not respectful of the loss of childcare"; "My institution enacted strict... "shift schedules" that were outside of childcare hours so I was unable to work a full work week. However, I was expected to produce the same (if not more) results/data to make up for the time we were locked out" (additional examples in *Table 1*).

### Impact of COVID-19 on international postdocs

When we focused on residency status, we saw that more international postdocs (vs. US citizens/PR) reported having food needs unmet (4% vs 1%), while US citizens/PR (vs. international postdocs) reported more difficulty in obtaining childcare (13% vs 9%) (*Figure 3A*). Additionally, international respondents (n=718) expressed specific worries regarding their residency status. The majority of international postdocs reported apprehension about immigration or visas either due to recent policy changes in the US (84%) or in general (11%) (*Figure 3B*). The primary concerns noted were traveling (75%), US immigration policy changes (69%), and travel bans (68%) (*Figure 3C*, *Table 1*). Furthermore, more international females than males were worried about immigration issues (89% vs 78%) (*Figure 3—figure supplement 1A*); specifically, travel (80% vs 70%), delays in visa renewal (65% vs 56%),

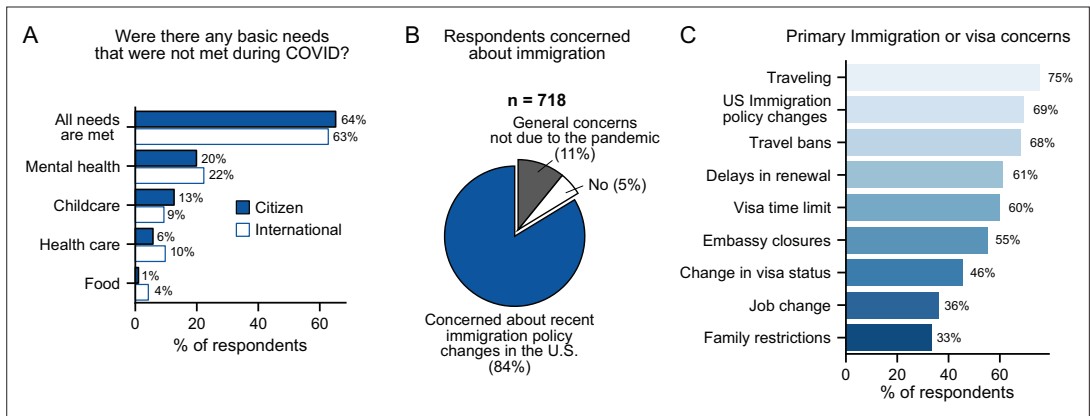

**Figure 3.** Impact of COVID-19 on international postdocs. (**A**) Citizenship status had a significant impact on childcare (multivariate logistic regression, P=0.027, OR=1.49 [95% CI; 1.05–2.13]) and food (multivariate logistic regression, P=0.002, OR=0.27 [95% CI; 0.11–0.62]) basic needs that were left unmet during the pandemic (n=1,657). (**B**) Most international postdocs (n=718) who were concerned about immigration and policy changes in the US said these were due to the pandemic. (**C**) The primary immigration or visa concerns of international postdocs (n=718). See *Figure 3—figure supplement 1* for breakdown of immigration concerns by gender, race and ethnicity, and LGBTQ status.

The online version of this article includes the following figure supplement(s) for figure 3:

**Figure supplement 1.** Immigration concerns broken down by demographic data.

and travel bans (72% vs 62%) (*Figure 3—figure supplement 1B*).

Compared to white international postdocs, both Asian and URM international postdocs were more concerned about changes in jobs (42% Asian and 43% URM vs 29% White), visa status (49% and 52% vs 41%) and US immigration policy (70% and 80% vs 66%); and were less concerned about traveling (62% and 78% vs 87%). Additionally, international Asian postdocs were also less concerned about Embassy closures (51% Asian vs 59% white) and travel bans (57% vs 78%) (*Figure 3—figure supplement 1C*). Finally, international LGBTQ postdocs were more concerned about US immigration policy changes compared to non-LGBTQ postdocs (80% vs 69%).

### Impact of the pandemic on mental health and wellness

Overall, 76% of respondents stated that the COVID-19 pandemic had impacted their mental health, with 32% stating that it had a high or very high impact (*Figure 4A*). In open-ended responses, postdocs mentioned significant impacts on their mental health due to isolation and pandemic associated stressors leading to reduced productivity, inability to focus and work effectively: "My mental health has been struggling, which has negative consequences on my ability to focus"; "The isolation has had a negative effect on my mental health … "; "Mental health diminished productivity despite being

able to work 100% remotely" (see *Table 1* for more examples).

All gender, race and ethnicity, and identity groups indicated a significant impact on mental health. However, certain groups reported more of an impact than others; females reported a greater impact than males (80% vs 68%); white and URM postdocs reported more of an impact than Asian postdocs (78% and 80% vs 68%); members of the LGBTQ community (83% vs 75%) and postdocs with disabilities (88% vs 76%) reported more of an impact than postdocs not identifying with these groups (*Figure 4B*).

### Effect of institutional support on mental health and wellness

Parallel to this impact on mental health, access to institutional mental health resources rose by 15% (*Figure 5A*), which appears to be linked to an increase in awareness, although only 17% of postdocs indicated use of these resources. Female postdocs reported higher usage of these resources compared to male postdocs (21% vs 10%; *Figure 4B*). Notably, postdocs without access to, or who were unaware of, institutional mental health resources were more likely to have their mental health impacted by COVID-19 than postdocs with access to those resources (*Figure 5B*).

Indeed, postdocs were more likely to have their mental health needs met if their institution provided these resources (84%) than if their

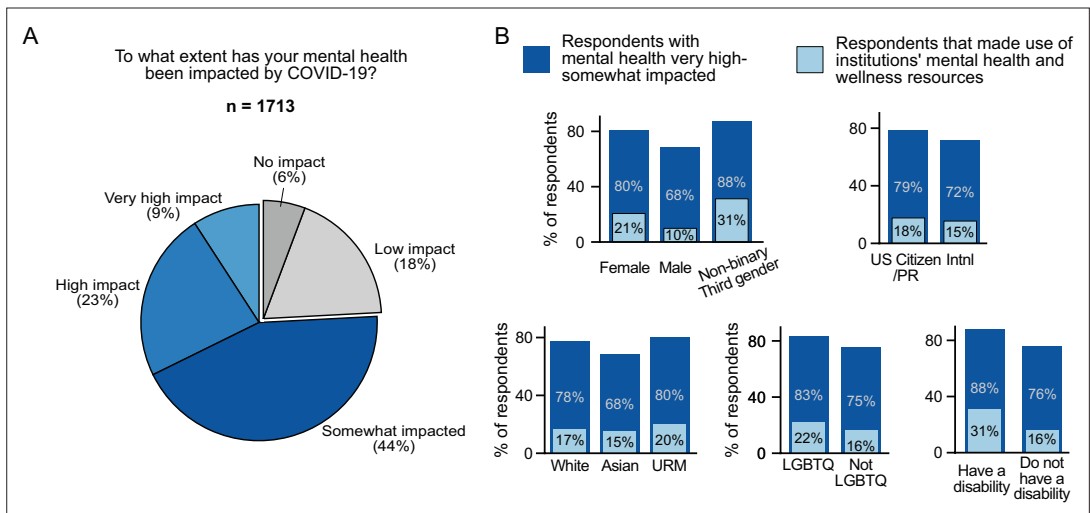

**Figure 4.** Impact of COVID-19 on mental health. (**A**) The majority of survey respondents stated that COVID-19 had impacted their mental health while only 6% stated that it had no impact (n=1,713). (**B**) Although most surveyed postdocs stated that their mental health was impacted (very high impact, high impact, and somewhat impacted), only a smaller percentage of postdocs utilized mental health and wellness resources provided by their institution. Females were impacted more than males (multivariate ordinal logistic regression $P=6.98 \times 10^{-11}$, OR=1.90, [95% CI; 1.57–2.30]; n=1,611) and used more institutional resources (multivariate ordinal logistic regression, $P=1.39 \times 10^{-7}$; OR=2.33 [95% CI; 1.70–3.20]; n=1,607). Asian postdocs were less impacted compared to white respondents (multivariate ordinal logistic regression, P=0.028, OR=0.76, [95% CI; 0.59–0.97]; n=1,611). Postdocs who identified as LGBTQ (multivariate ordinal logistic regression, $P=6 \times 10^{-4}$, OR=1.84 [95% CI; 1.30–2.60]; n=1,611) and postdocs with disabilities (multivariate ordinal logistic regression, P=0.0053, OR=2.26 [95% CI; 1.27–4.01]; n=1,611) also reported higher impact on their mental health.

institution either did not provide them (42%) or if they were unaware of these resources at their institution (68%, *Figure 5C*). Postdocs at institutions that provided mental health resources were also more likely to have all their basic needs met (69%) compared to those without (35%) or unaware of these resources (50%) (*Figure 5D*). Unsurprisingly, postdocs that did not have access to mental health resources at their institutions, were also more likely to have other basic needs unmet such as food (8% (no) vs 2% (yes)) or health care (21% (no) vs 7% (yes); *Figure 5—figure supplement 1A*).

In written responses, several postdocs mentioned that their institutions provided mental health resources; however, they were often unaffordable or inaccessible: " … doesn't take postdocs appointments for mental health or other such services they are completely booked [*sic*]."; "Limited financial resources to pay to access mental health resources as "free" sessions through employer was used pre-COVID."; " … has mental health resources but they are not free at all."; "Note, the mental health resources available to post-docs and faculty here are minimal, but they do exist -- mostly things like meditation workshops. … However, whether any of these

resources are available to postdocs depends on whether our funding is internal ('associates', as I am) or external ('fellows', who receive fewer benefits)". These stark differences between institutions with mental health resources and those without, highlight the widespread importance of mental health care and its correlation with quality of life in the postdoctoral population.

As previously indicated (*Figure 2F*)**,** access to a PDA and/or a PDO also increased the likelihood of mental health needs being met. This trend may be due in part to a larger proportion of postdocs with access to a PDO/PDA also having access to mental health resources (82% and 80%) compared to those that did not (59% and 61%) or were unaware (66% and 60%) (*Figure 5E–F*). Postdocs with a PDO/PDA were also more likely to use their institution's mental health resources (19% and 18%) compared to those that did not have access (9% and 11%) or were unaware of these resources (13% and 9%, *Figure 5E–F*).

### How the pandemic affected the career trajectory of postdocs
The pandemic dramatically impacted career trajectories of the postdocs due to lab shutdowns, inability to communicate with faculty supervisors

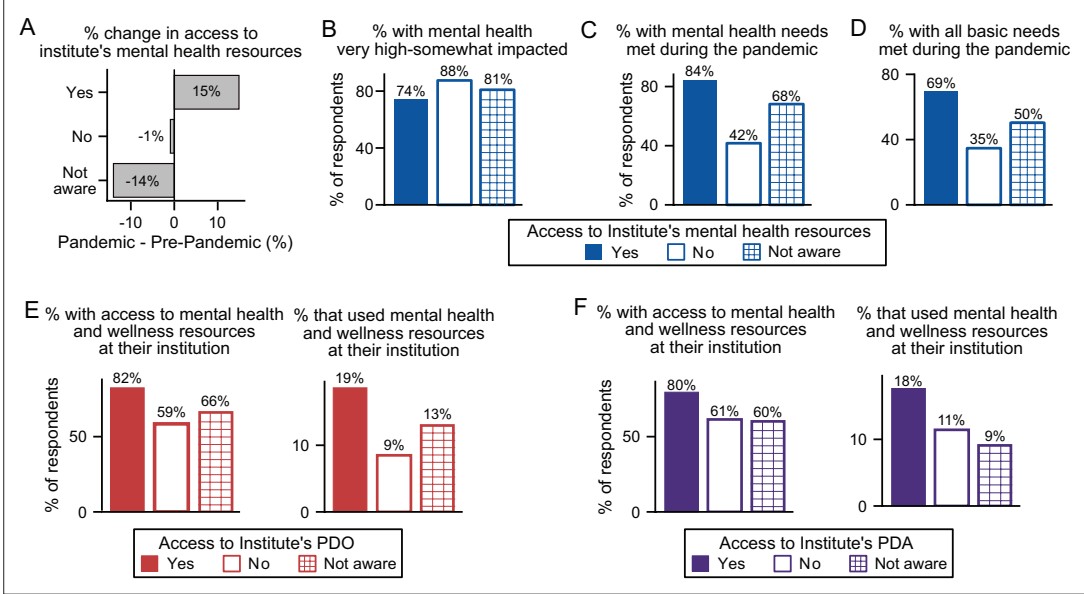

**Figure 5.** The effect of institutional resources on having mental health needs met. (**A**) During the pandemic, more individuals had access to mental health resources, which was reflected in an increased awareness of these resources available at their institution (multivariate logistic regression, $P=0.038$, OR=1.35 [95% CI; 1.02–1.78]; n=7,047). That increase in awareness is proportional to the increase in respondents stating that their institution has available mental health resources. (**B**) Not having access (multivariate ordinal logistic regression, $P=4.3 \times 10^{-6}$, OR=3.04, [95% CI; 1.89–4.88]), or not being aware (multivariate ordinal logistic regression, $P=8.52 \times 10^{-4}$, OR=1.50, [95% CI; 1.18–1.91]) of mental health resources increased mental health impact during the pandemic (n=1,608). (**C**) A larger portion of postdocs who did not have access to (multivariate logistic regression, $P=1.03 \times 10^{-13}$, OR=0.13 [95% CI; 0.08–0.22]) or were unaware of (multivariate logistic regression, $P=6.79 \times 10^{-10}$, OR=0.39, [95% CI; 0.29–0.52]) mental health resources reported having their mental health basic needs met (n=1,610) compared to postdocs who had access to mental health resources. (**D**) A smaller portion of postdocs who did not have access to (multivariate logistic regression, $P=5.05 \times 10^{-8}$, OR=0.22 [95% CI; 0.13–0.38]) or were unaware of (multivariate logistic regression, $P=6.12 \times 10^{-9}$, OR=0.45, [95% CI; 0.35–0.59]) mental health resources reported having all their basic needs met (n=1,610). See ***Figure 5—figure supplement 1A*** for other basic needs unmet vs access to mental health resources. (**E–F**) Having a PDO or a PDA increased access to mental health resources (PDO (multinomial logistic regression, $P=1.36 \times 10^{-5}$, OR=4.61 [95% CI; 1.86–10.58]; n=1,610); PDA (multinomial logistic regression, $P=0.0073$, OR=2.94 [95% CI; 1.34–6.47]; n=1,610)), whereas only access to PDOs increased the use of mental health resources (PDO (multinomial logistic regression, $P=0.035$, OR=2.19 [95% CI; 1.06–4.53]; n=1,607)).

The online version of this article includes the following figure supplement(s) for figure 5:

**Figure supplement 1.** Relationship between having access to mental health resources and having basic needs met.

and research group members, and most significantly, additional family responsibilities, etc., compared to one year earlier (see word cloud in ***Figure 2A–B*** and select comments in ***Table 1***). This resulted in reduced research productivity, delayed job searches, lowered confidence in attaining the desired career, and uncertainty in overall career trajectory. Even though the postdocs were older and had more years of experience when re-surveyed (***Figure 1D–E***), a smaller proportion were currently looking for positions (64% pre-pandemic, 56% during the pandemic), with 11% of postdocs specifically delaying their job search because of the pandemic (***Figure 6A***). In addition, postdocs were less confident in

achieving their career goals than before the pandemic (***Figure 6B***, 76% said they were very confident to somewhat confident pre-pandemic compared to 68% during the pandemic), which may be contributing to the observed decline in those actively pursuing new positions (***Figure 6A***). Furthermore, more postdocs were undecided about their future careers than before the pandemic (9%–12%) (***Figure 6C***). Together, these results highlight the substantial increase in career uncertainty felt by postdocs.

Overall, 34% of postdocs reported changing their career plans during the pandemic, with 23% of respondents indicating that COVID-19 was the direct cause of their change (***Figure 6D***).

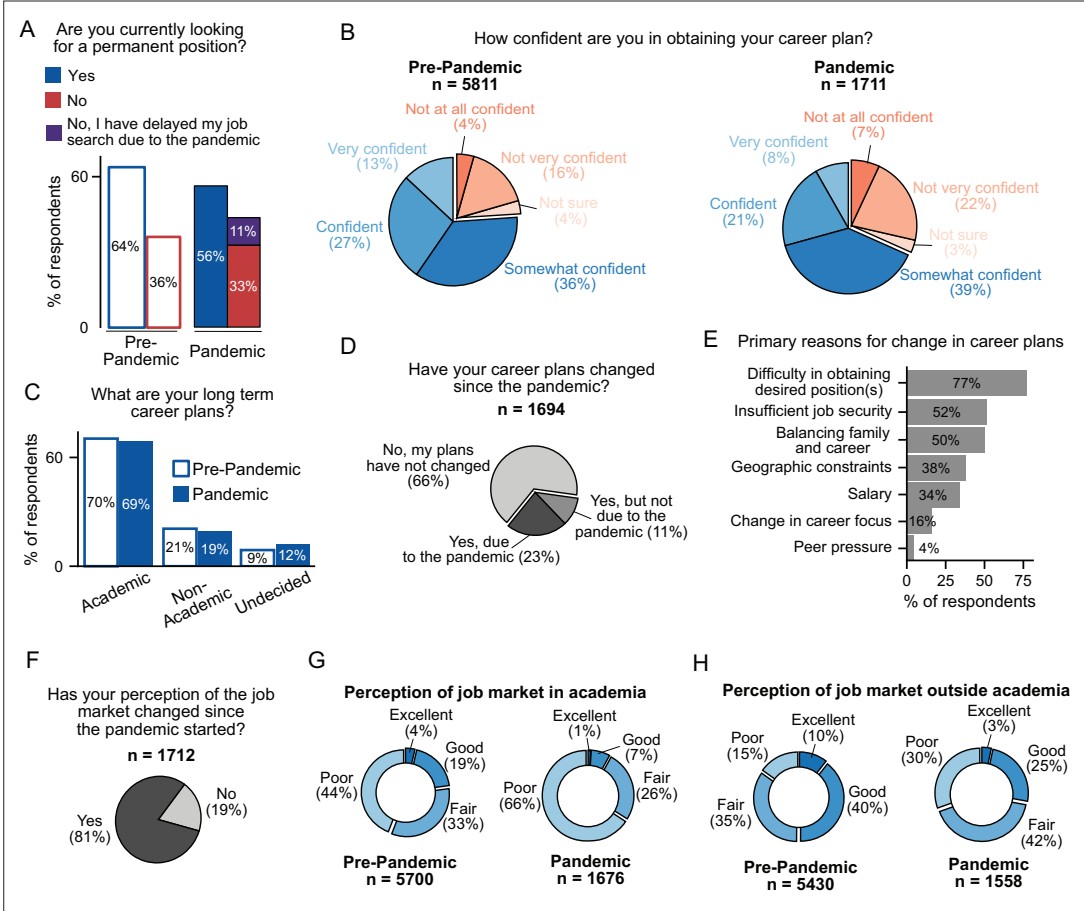

**Figure 6.** The effect of COVID-19 on the career trajectories of postdocs. (**A**) Fewer postdocs are actively looking for a permanent position than before the pandemic (multivariate logistic regression, $P=8.24 \times 10^{-7}$, OR=0.75 [95% CI; 0.67–0.84]; n=6,899). See *Figure 6—figure supplement 1A* for breakdown by type of position. (**B**) Postdocs are less confident in their ability to obtain their desired career than before the pandemic (multivariate ordinal logistic regression, $P=2.39 \times 10^{-19}$, OR=0.62 [95% CI; 0.56–0.69]; n=6,964). (**C**) The long-term goals of postdocs have not shifted during the pandemic. However, a larger proportion of postdocs are now uncertain about their career trajectories (multinomial logistic regression, $P=0.0073$, OR=1.28 [95% CI; 1.07–1.54]; n=6,954). (**D**) 34% of postdocs indicated that their career plans changed since the pandemic started (n=1,694). (**E**) Primary reasons for changes in career trajectory (n=388). See *Figure 6—figure supplement 1B* for breakdown by citizenship status. (**F**) During the pandemic, the perception of both the academic and non-academic job markets has declined (n=1,712). See *Figure 6—figure supplement 1C* for breakdown by citizenship status. (**G**) A decrease in the perception of the job market both in academia (multivariate ordinal logistic regression $P=2.51 \times 10^{-55}$, OR=0.39 [95% CI; 0.35–0.44]; n=6,870) and (**H**) outside academia (multivariate ordinal logistic regression, $P=6.94 \times 10^{-68}$, OR=0.38 [95% CI; 0.34–0.42] n=6,513) was observed during the pandemic compared to the pre-pandemic survey.

The online version of this article includes the following figure supplement(s) for figure 6:

**Figure supplement 1.** Change in career plans broken down by demographics.

The postdocs that changed their career plans (both due to and not due to the pandemic), were more likely to be undecided about future careers (20% (due to the pandemic); 20% (not due to the pandemic) vs. 7% (did not change career plans)) or considering non-academic positions (28%; 37% vs 14%), and much less likely to be seeking an academic position (51%; 43% vs 79%) compared to postdocs who did not change their career plans (66% of surveyed postdocs) (*Figure 6—figure supplement 1A*).

Of those postdocs who indicated a change in career plans (both due and not due to the pandemic), the main reasons cited were: (i) difficulty in obtaining the desired position (77%), (ii) insufficient job security (52%), and (iii) balancing family and career (50%) (*Figure 6E*). Additionally, reasons for career change differed by citizenship

status and race/ethnicity. International post-docs cited more peer pressure (8% vs 1%) and geographic constraints (42% vs 35%) than US citizen/PR, while the latter noted more difficulty in obtaining desired positions (83% vs 69%) as well as balancing family and career (58% vs 39%, *Figure 6—figure supplement 1B*). More white and URM postdocs indicated geographic constraints as a reason for changing career trajectory compared to Asians (41% vs 29%, *Figure 6—figure supplement 1C*). Lastly, no differences were observed by gender or identity groups with respect to reasons for changing career trajectory (multivariate logistic regression, *P*>0.05, data not shown).

The majority of postdocs surveyed also reported a change in their perception of the job market (81%) (*Figure 6F*), with more US citizens/PR than international postdocs (85% vs 74%, *Figure 6—figure supplement 1D*) reporting a change in perception. No differences were observed based on gender or identity groups (multivariate logistic regression, *P*>0.05, data not shown). This altered perception was observed for both the academic and non-academic job markets (*Figure 6G and H*). Overall, the majority of the respondents viewed the current academic job market as poor (66%) or fair (26%), which is a significant change compared to the pre-pandemic survey, where fewer postdocs viewed the market as poor (44%) and more viewed it as fair (33%). Although the perception of the job market outside of academia was better - 28% of the respondents found it either excellent or good compared to academic careers (8%) - there was still a decrease in perception from the pre-pandemic survey.

### Career changes during the pandemic

The postdoctoral position is considered temporary, with the ultimate goal of providing the necessary training and experience to successfully transition to more permanent careers. To better understand the effects of the pandemic on career outcomes, we surveyed those who were no longer in postdoctoral positions. Of those who responded to the second survey, 11% (219/1,941) were no longer postdocs, with 14% indicating that this career transition was a consequence of the pandemic (*Figure 7A*).

Overall, 56% of the postdocs who made career transitions remained in academic positions (clinical, research staff, or faculty), while nearly 8% were unemployed. When we separately examined postdocs who made career transitions as

a consequence or irrespective of the pandemic, we observed a profound difference in career outcomes. The former group was more likely to be unemployed (38% vs 6%) and less likely to be in academic positions than postdocs who chose to leave their position regardless of the pandemic (24% vs 65%). However, we observed little difference in those pursuing non-academic careers (38% vs 29%; *Figure 7B*).

## Discussion

Early in March of 2020, the COVID-19 pandemic forced research facilities across the US to drastically alter their activities. This resulted in a cascade of events, including loss of research progress, delayed career advancement and a significant disruption of work and life activities. Although there have been multiple reports of the pandemic's impact on the STEM workforce, few have discussed the impact on postdocs specifically (*Woolston, 2020a*; *Woolston, 2020b*; *Gao et al., 2021*; *Doyle et al., 2021*; *Korbel and Stegle, 2020*; *Carr et al., 2021*; *Aubry et al., 2021*; *Seitz et al., 2020*; *Deryugina et al., 2021*; *Nature, 2020*).

To investigate the impact of the pandemic on the postdoctoral experience, we took advantage of our recently completed 2019 National Postdoctoral Survey 2.0 and re-surveyed the same population during the pandemic. Rather than interrogating during the first few months with all the uncertainty of full lockdowns, etc., we surveyed postdocs at the start of the second wave (in the US), when many institutions were partially open with social distancing, masking, etc., but still before access to vaccines, and immediately before the 2020 US elections. The data provided a unique opportunity to directly assess the effects of the pandemic on the post-doctoral experience and perform a comparative analysis of the same cohort before and during the pandemic. As well, the pandemic survey was only open during a restricted period (1 month), and therefore allowed us to capture a defined period of the pandemic.

The results of the pandemic survey showed that access to institutional resources like PDAs and PDOs is important not only for postdocs to complete their work in safe and supportive environments, as is often the focus of these institutional assets, but also for their mental and physical wellbeing. This intriguing correlation between postdocs' satisfaction with their institution's response to the pandemic, specifically whether their mental health, childcare and basic

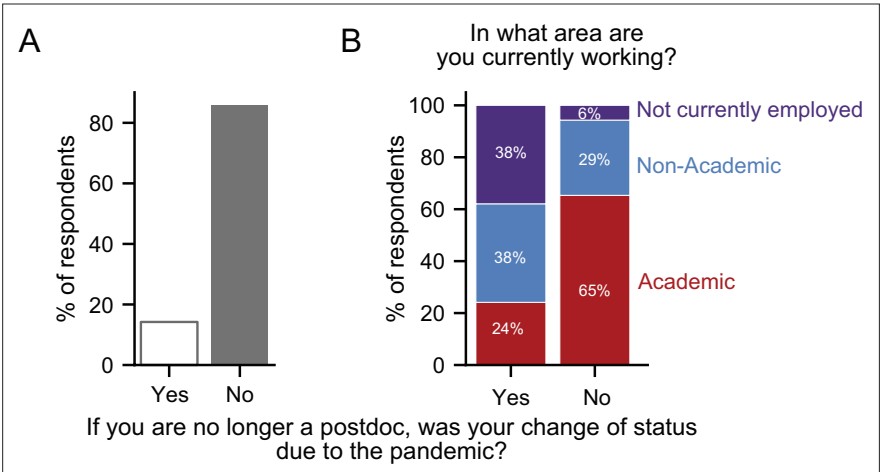

**Figure 7.** Career transitions made during the pandemic. (**A**) 14% of respondents who indicated that they are no longer a postdoc, stated that their transition was a consequence of the pandemic (n=218). (**B**) Postdocs who transitioned due to the pandemic were more likely to be unemployed (purple) and less likely to have an academic position (red) than postdocs whose transition was not a consequence of the pandemic (Chi-squared test, P=6.69 × 10⁻⁸, $\chi 2$=33.04; n=205).

needs were met, are important aspects that need further examination. In the ~7 months between the beginning of the pandemic and this survey, potential long-term consequences impacting career progression for many postdocs, such as delayed job searches, lost productivity, canceled positions, fewer opportunities and altered career trajectories have already been observed. These experiences may result in more postdocs becoming disenchanted with their career trajectories.

Our results have been implicated in other studies, including a Nature survey early in the pandemic, that found postdocs all over the world were impacted by loss of productivity, uncertain job prospects, lab closures, travel bans and experienced mental health issues (*Woolston, 2020a*; *Woolston, 2020b*). In more recent reports, mental health and career impacts due to the pandemic have been experienced by all members of the university, across a multitude of fields (*Doyle et al., 2021*; *Gao et al., 2021*; *Korbel and Stegle, 2020*; *Servick et al., 2020*; *Yan, 2020*). Unfortunately, there has been a greater impact on underrepresented minorities and female academics, especially those with young children (*Staniscuaski et al., 2021*; *Deryugina et al., 2021*). These reports and ours highlight the pandemic's toll on all members of the academy, and emphasize where institutional resources should be targeted.

The longer-term effects of the pandemic on the postdoctoral population, however, are not able to

be predicted from our data, since our study was only conducted over a short period of time during the pandemic. Thus, further studies are needed to assess how the pandemic ultimately affects career trajectories. Likewise, it is also premature to predict irreversible trends. However, while we are barely out of the COVID-induced disruptions, there are some glimpses of positive changes that are emerging (*Gardner et al., 2021*; *Gould, 2021*; *Shah et al., 2021*). For example, in order to mitigate isolation and perpetuate a learning environment when labs shut down in early 2020, institutional entities responsible for professional skill-building and career-promoting programming pivoted to remote platforms with some interesting outcomes (*Andrade et al., 2022*). Remote sessions now appear to be not only acceptable, but may even be preferred by trainees, as they are more accessible, provide more efficient use of time and offer flexibility (such as having the possibility to watch recorded sessions later). Since PDOs and PDAs often provide these types of professional development programs, this may explain why postdocs with access to these institutional assets were significantly more satisfied. Again, this is a potentially important factor that needs further exploration.

### Limitations and future directions
As with all such surveys, there were a few limitations to our study. First, while the pandemic survey was conducted on a subset of the pre-pandemic respondents, the responses were collected anonymously according to our IRB protocol. We therefore were unable to do a direct one-to-one comparison of pre-pandemic to pandemic responses on an individual basis. Second, in the pandemic survey, we did not directly ask if respondents were parents or caregivers and were only able to partially assess this through questions that required written responses to pandemic related stressors. Therefore, we were limited in our ability to assess exactly how many postdocs had caregiving responsibilities that disrupted their research and career progression, and further studies are needed to investigate how wide-spread the impact was on these postdocs.

Third, because of small sample sizes, we were also limited in evaluating certain demographic metrics such as LGBTQ and disability status, as well as individuals who identified as non-binary/third gender. Furthermore, to parse out potential differences between racial and ethnic groups, we pooled individuals into three broad groups; white, Asian and URM (see Methods for more

details about how these groups were assigned). As a result, specific races/ethnicities were not examined individually in this study.

Individual pandemic experiences varied greatly throughout the pandemic, undoubtedly influenced by factors such as geographic location, type of institution, timeframe within the pandemic, caregiving responsibilities, and how amendable a person's research is to remote work. Our study did not assess how effects differed across the country based on geographic location or type of institution (for example, an institute with more resources, budget, and personnel dedicated to postdoc affairs vs. others). Nevertheless, we did capture responses from postdocs across the US, with higher response rates in locations that would be expected to have more postdocs, such as on the East Coast and in California (*Figure 1—figure supplement 1A-B*). We also did not directly analyze how the experience varied for postdocs in different fields. Some postdocs were likely able to transition readily to remote work for a period of time, with little to no disruption to their research (e.g., biostatisticians), while others may have had their research productivity significantly hindered or disrupted due to limited access to their laboratories. These are all important factors that will need to be evaluated in future studies.

Lastly, our study focuses on a specific period of time (Fall 2020) when the urgency and uncertainty of the initial shutdown period had lessened. However, there have been successive waves of the pandemic, the effects of which will require additional investigation. Further studies should also ascertain whether the pandemic had any positive impacts on postdocs, such as the ability to continue to work remotely under certain circumstances or make new professional connections via video conferencing, etc. (*Levine et al., 2021*; *Gardner et al., 2021*; *Gould, 2021*; *Shah et al., 2021*).

## Conclusion

As we and others have previously reported, postdocs are an often overlooked and forgotten population in academia, with a non-negligible number of institutions being unaware of their total postdoc population, let alone the concerns of postdocs. Access to employee benefits varies widely, with untoward consequences for postdocs who lack access to certain essential services (*National Research Council, 1969*; *Shen, 2015*; *Alund et al., 2020*). In a position that emphasizes sacrifice for research, institutions should be more proactive in discerning the basic needs of their postdocs, and make appropriate resources and benefits available, including access to paid family leave, healthcare, mental health and wellness services for the postdocs and their families, as well as affordable childcare options.

PDOs are the appropriate institutional entities to ensure that postdocs' minimal basic needs are met and should assume responsibility for developing policies, disseminating essential benefits and coordinating with their PDAs in providing career-enhancing programming. Furthermore, having postdoctoral leadership and organization, through PDAs, gives postdocs a voice and an avenue for communicating the needs of postdocs within the institution. Moving forward, we plan to continue to survey this cohort of US-based postdocs in order to generate a better understanding of the long-term consequences of the COVID-19 pandemic on postdoc experiences and outcomes. Ultimately, understanding the needs of this critical workforce will also broadly benefit the future of science and research.

## Methods

### Survey design and dissemination

The National Postdoctoral Survey was designed to capture the experiences and demographic information of postdoctoral fellows and scholars across the US. The survey was initially conceived and developed by postdocs within the University of Chicago's Biological Sciences Division Postdoctoral Association (PDA) in 2016, in order to identify important issues within the postdoctoral community and inform and equip those who advocate for postdoctoral policies to make positive changes. The results of the first National Postdoc Survey were published by McConnell et al. in 2018.

In 2019, a second updated version of the National Postdoc Survey was launched by the University of Chicago PDA. This version, referred to as the "pre-pandemic survey", collected responses from postdocs in the US from June 4, 2019 until December 31, 2019. Our survey design and dissemination protocol were approved by the University of Chicago Institutional Review Board, IRB Protocol Number 15–1724.

In order to make postdocs across the US aware of the survey, multiple types of grassroots outreach were used in a similar manner to McConnell et al. First, we performed online website searches for Postdoc Offices (PDOs) at doctoral degree-granting universities or

research institutions in the US that train post-docs. We compiled a list of publicly available email addresses for institutional representatives of these PDOs. If we were unable to identify a PDO, or if an institution did not have a PDO, then contact information was collected instead for an administrative or faculty representative within an Office of Research, Graduate School, or Provost, or for a similar official who might have access to postdocs. We also collected contact information, if available, for postdoc leaders of Postdoc-toral Associations (PDA), who were contacted if institutions did respond to our initial outreach or if an institution's response rate was deemed low compared to the 2016 National Postdoc Survey. We emailed over 400 institutional PDOs, other administrative contacts or PDA leaders, described the goals of the survey, and asked them to distribute our survey link and invitation to the postdocs at their institution. Over the course of the seven months that the survey was open, follow-up emails were sent to our contacts to remind them to send the email to their post-docs, or to distribute the survey link if they had not already done so.

In addition to our outreach to institutional representatives, we shared the survey on social media websites including Twitter and LinkedIn, launched a website dedicated to the National Postdoc Survey, and prepared an email campaign to advertise the survey which was distributed by the National Postdoctoral Association to its large national listserv of postdocs and postdoc advocates. These additional methods were used to enhance awareness of the survey and distribute the survey link directly to postdocs who may not have received it through their institution.

During the seven months that the survey was open, responses from 6,292 postdocs were collected from over 300 institutions in nearly every state in the nation. All responses were collected anonymously, but most respondents (5,594 (89%)) voluntarily provided contact infor-mation in a separate form to draw names for survey incentive prizes. Of the 6,292 respondents to the survey, 5,929 identified as postdocs at a US institution and only their responses were used for analysis.

While analysis of the 2019 pre-pandemic survey data was underway, the COVID-19 pandemic commenced, and it became evident that a follow-up survey was necessary to assess the changes brought on by the pandemic in the mindsets and current situations of postdocs. Questions were designed in 2020 for a shorter "pandemic survey" to query what changes the

postdocs experienced in their career goals and whether their plans changed since the pandemic started, current perceptions of the job market in academia, and how their research and life has been affected by the pandemic. All postdocs who completed the initial pre-pandemic survey and submitted their email addresses for recon-tact (5,594 in total) were asked to complete this second pandemic survey, which was launched on October 1, 2020 and was only open for one month. Many emails bounced, suggesting that many postdocs from the previous cohort had defunct addresses.

In total, 1,942 responses to the pandemic survey were collected. Of these responses, 1,722 (88.6%) were submitted by researchers currently in postdoctoral positions in the US, and these responses are analyzed here. Pre-pandemic and pandemic survey questionnaires are included in *supplementary file 1* and *supplementary file 2* respectively.

### Data analyses

Participants were queried about their race and ethnicity with the following options: white/Cauca-sian, Asian/Asian American, South Asian/South East Asian, Black/African American, Hispanic/Latino, Middle Eastern, Native American or Alaska Native, and Pacific Islander or Hawaiian Native. For all analyses, due to low numbers of respondents, certain racial/ethnic categories were consolidated into three groups: (1) underrepre-sented minorities (URM) as defined by the NIH (https://diversity.nih.gov/about-us/population-underrepresented): Black or African American, Hispanic or Latino, Native American or Alaska Native, Native Hawaiian or Pacific Islander; (2), Asians (Asian/Asian American and South Asian/Southeast Asian); and (3) white (white/Caucasian and Middle Eastern – which were grouped based on the categorization provided by the NIH and US census). The NIH also defines those with disabil-ities (analyzed separately here) and those from disadvantaged backgrounds (this category was not queried in this study) to also be underrepre-sented in the biomedical, clinical, behavioral, and social sciences. Unless stated otherwise, interna-tional postdocs and US citizens/permanent resi-dents were included in all the analyses as long as the postdocs were working in the US.

All questions were optional, thus, non-respondents for each question were removed before each analysis. The number of respondents used for each analysis are indicated in the legend/figure panels. To assess similarities between the

pre-pandemic and pandemic surveys, we used the Chi-square test in the presence of categorical data (basic R function). We used the same test to assess the postdocs who transitioned out of their position during the pandemic, due to the lack of additional data available for this group. For the other analyses, we performed multivariate tests including gender, race and ethnicity, residency, LGBTQ status, having a disability, access to a PDO and access to a PDA as covariates. To assess differences, we used either ordinal logistic regression in the presence of ordinal dependent variables (using the R package "MASS"), logistic regression (basic R function) and multinomial logistic regression (using R package "nnet") accordingly. Results are described in the legend as Odds Ratio (OR) [95% Confidence Interval (CI)] with the p-value. We considered p-values <0.05 to be significant. In the manuscript, p-values of <0.05 were identified as *, $P<0.01$ ** and $P<0.001$ ***. Word clouds were generated in Python using the wordcloud package. Figures were generated using Python version 3.7.6.

### Acknowledgements
We thank Dr. Valerie Miller (UIC) for her assistance with the 2019 survey instrument design and dissemination; the Future of Research directors for suggesting the inclusion of mental health and wellness queries in the survey; and we are especially grateful to Drs. Erin Heckler (Yale University) and Imogen Hurley (UW Madison) for valuable comments. We are very grateful to the many Postdoctoral Associations, administrators, faculty directors, and others who helped distribute our survey to postdocs across the country. Finally, we wish to especially express our gratitude to the postdocs who shared their experiences with us by participating in the surveys.

**Andréanne Morin** is in the Department of Human Genetics, University of Chicago, Chicago, United States

amorin@uchicago.edu

 http://orcid.org/0000-0002-5535-9518

**Britney A Helling** is in the Department of Human Genetics, University of Chicago, Chicago, United States

 http://orcid.org/0000-0002-3895-5107

**Seetha Krishnan** is in the Department of Neurobiology and Institute for Neuroscience, University of Chicago, Chicago, United States

 http://orcid.org/0000-0002-6218-3995

**Laurie E Risner** is in the Department of Pediatrics, University of Chicago, Chicago, United States

 http://orcid.org/0000-0002-1954-6623

**Nykia D Walker** is in the Ben May Department for Cancer Research, University of Chicago, Chicago, United States. Current institution: University of Maryland at Baltimore County, Baltimore, Maryland, United States

 http://orcid.org/0000-0001-7603-9276

**Nancy B Schwartz** is in the Department of Pediatrics and the Department of Biochemistry and Molecular Biology, University of Chicago, Chicago, United States

n-schwartz@uchicago.edu

 http://orcid.org/0000-0002-7613-0894

*Author contributions:* Andréanne Morin, Conceptualization, Software, Formal analysis, Validation, Investigation, Visualization, Methodology, Writing – original draft, Writing – review and editing; Britney A Helling, Conceptualization, Formal analysis, Methodology, Writing – original draft, Writing – review and editing; Seetha Krishnan, Conceptualization, Software, Formal analysis, Validation, Investigation, Visualization, Methodology, Writing – original draft, Writing – review and editing; Laurie E Risner, Conceptualization, Methodology, Writing – original draft, Project administration, Writing – review and editing; Nykia D Walker, Writing – review and editing; Nancy B Schwartz, Conceptualization, Supervision, Investigation, Visualization, Methodology, Writing – original draft, Project administration, Writing – review and editing

*Competing interests:* The authors declare that no competing interests exist.

*Ethics:* Participation in this survey was completely voluntary. In the introduction to this survey, we informed the participants of its purpose, and that results of the survey would be disseminated, in aggregate. All responses were recorded in a secure RedCap Database, so they could not be traced back to individual respondents. Responses were combined for data analysis to maintain respondent anonymity throughout data analysis. Our survey design and dissemination protocol was approved by the University of Chicago Institutional Review Board, IRB Protocol Number 15-1724.

### Funding

| Funder | Grant reference number | Author |
| --- | --- | --- |
| Fond de recherche du Quebec en Sante | Postdoctoral fellowship | Andréanne Morin |
| National Institute on Drug Abuse | Grant Number T32DA043469 | Seetha Krishnan |

The funders had no role in study design, data collection and interpretation, or the decision to submit the work for publication.

**Decision letter and Author response**
Decision letter https://doi.org/10.7554/eLife.75705.sa1
Author response https://doi.org/10.7554/eLife.75705.sa2

# Additional files

## Supplementary files
• Supplementary file 1. Pre-pandemic survey questionnaire.
• Supplementary file 2. Pandemic survey questionnaire.
• Supplementary file 3. Race and ethnicity distribution among respondents of the pre-pandemic and pandemic survey.
• Transparent reporting form

## Data availability
The survey questions can be found in additional data files (Supplementary File 1 and Supplementary File 2). All the statistical details and anonymized data such as the percentage of respondents can be found in figures, figure legends as well as method and results sections. Because of the sensitive information, the full data cannot be shared to maintain the confidentiality of surveyed subjects, personal information is not shared as stipulated in IRB.

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
