## [Decision Letter]

Thank you for submitting your article "The Effect of COVID-19 on the Postdoctoral Experience: a comparison of pre-pandemic and pandemic surveys" to eLife for consideration as a Feature Article. Your article has been reviewed by three peer reviewers, and the evaluation has been overseen by two members of the eLife Features Team (Julia Deathridge and Peter Rodgers). None of the reviewers wished to reveal their identity.

The reviewers and editors have discussed the reviews and we have drafted this decision letter to help you prepare a revised submission.

Summary:

This study investigates the impact of the COVID-19 pandemic on postdocs in the United States. Schwartz and colleagues surveyed 1,722 postdocs across 300 institutes in 2020 (during the second wave of the pandemic), and compared the results to a previous survey they conducted in 2019, which includes responses from almost 6,000 postdocs. The data revealed that the pandemic had widespread impacts on the mental health, research productivity, and career trajectories of postdocs. Particular demographic subgroups also faced additional burdens. Some of these challenges were less prevalent for respondents who had access to a Postdoctoral Office or Postdoctoral Affairs Association at their institution. However, there are a number of points that need to be addressed to make the article suitable for publication.

Essential revisions:

1) The results section is very detailed which makes it difficult to identify the main findings of the survey. To make the results easier to follow, please can you split this part of the manuscript into more sections (particularly the part on COVID-19 impact: lines 119-219) and give each subsection a heading summarizing the finding that is the most important. For example,

– lines 136-152 could fall under the heading "Postdocs with access to PDOs were more satisfied with their institution's response to the pandemic"

– lines 251-281: "Postdocs with access to mental health resources were more likely to have their basic needs met"

– Lines 284-298: "Postdocs were less confident in achieving their career goals than before the pandemic"

This may require changing the order in which some of the results are discussed so that those reporting on a related topic are more clearly grouped together. For example, all the data about the benefit of PDOs/PDAs could be grouped into a single section, and the data on basic needs (lines 153-162 and 209-219) could also be grouped together.

The above examples are not exhaustive and are just suggestions. Please use your own judgment to decide which findings to group together and the main result you would like to highlight in the relevant subheading.

2) The discussion frequently restates the results rather than interprets them (for example: Page 17, Lines 351 to 357; Page 18, Lines 375-378, 379-380; and Page 19, Lines 398-403, 406-407).

Please reduce these repetitions and expand on the points listed below:

i) As pointed out in line 364, there have now been a few reports about the pandemic's effect on the workforce, including postdocs. How does the data fit with this literature, and altogether what is the takeaway message with respect to the new findings presented here? How do you interpret the demographic differences observed?

ii) How the challenges faced by postdocs compare to the rest of academia.

iii) In Line 409 - Can you be more specific about "hints of long-term consequences"? What are you suggesting would be the long-term consequences of reduced productivity, delayed job searches, etc?

iv) Do your data speak to irreversible trends that will impact the workforce, or rather do they suggest areas where specific interventions could be impactful/warranted? What are the future directions of this work?

v) Were there any positive impacts of the pandemic?

3) Please add a Limitations section which should include the following points (as well as lines 379-393 in the Discussion):

i) The pandemic was experienced very differently and at different times among different geographic locations in the United States. There is no mention of this or how it may affect the conclusions of the manuscript since this is not a nationally representative sample of postdocs.

ii) The impact of the pandemic was different for postdocs in different fields. For example, basic scientists who had their labs shut down were in very different situations than biostatisticians who could more easily continue to work from home

iii) Potential for response bias. Because the pandemic survey was only sent to those who provided email addresses, it raises concerns that the people who responded were inherently different than those who did not provide an email address. Additionally, people who were unemployed may have been more likely to spend time answering the pandemic survey than those who were employed

iv) A main take home that the authors point out is that the presence of Postdoc Offices or Postdoc Associations as a positive impactor on some challenges faced by responders. However, based on the current study this claim is only correlative. For example, it is unclear if the respondents had accessed their local PDO/PDA or had just heard of them. Furthermore, various other contributors are also not included in the present study, such as type of institution (availability or abundance of resources, personnel dedicated to postdoc affairs, budgets, etc.), geographical location of the institution, number of postdocs within the institute, location of PDO (university-wide or college-specific), disciplinary area of research, and community and camaraderie among colleagues.

4) The article concludes that the pandemic profoundly impacted the career trajectories of postdocs. However, the long-term impact of the pandemic on the career trajectories of postdocs is not clear from the current data, since the survey results only represent a single snapshot in time during the first year of the pandemic.

Please can you tone down this statement and include the caveat that the current data only represents a snapshot in time. Furthermore, are there any other studies that support this claim?

5) Please update the definition for underrepresented that is used in this manuscript and analyses (pages 22-23) so it is consistent with the NIH definition: https://diversity.nih.gov/about-us/population-underrepresented.

6) The paper contains a number of univariate analyses. Please can you indicate whether or not you adjusted your p-values for multiple comparisons?

7) Figure 1 - figure supplement 1: In C, D, E the bar heights don't match the number labels (%)- e.g. all bars that say 1% should be the same height, bars that say 2% should be the same height, bars that say 0% should be the same height, bars that say 4% should be the same height. The bar marked 8% in panel D doesn't match where 8% would be on the y axis, etc. Please double-check the data and replot or relabel as appropriate.

8) Your analyses could be strengthened by using multivariate ordinal logistic regression models and/or logistic regression models. On page 9, line 168, you state that the majority of Asian post-docs were international and more likely to have unmet healthcare needs. If you created a multivariate logistic regression model, the reader would be able to determine whether race or international status (and other factors) were independently associated with unmet needs. Please consider doing this for some or all of the analyses; however, we do not insist on this.

9) Fig 5D/Supplemental Fig 5: it would be interesting to see what these data look like by gender, particularly as it appears to be a general national trend that women are leaving the workforce at higher rates and experiencing particular challenges balancing family and career. Again, please consider doing this analysis; however, we do not insist on this.

---

## [Author Response]

[Editor’s note: Further changes were made following revisions to the manuscript in response to queries and discussions with the editor. As a result, some of the comments below may differ from the published version of the article.]

Essential revisions:1) The results section is very detailed which makes it difficult to identify the main findings of the survey. To make the results easier to follow, please can you split this part of the manuscript into more sections (particularly the part on COVID-19 impact: lines 119-219) and give each subsection a heading summarizing the finding that is the most important. For example,– lines 136-152 could fall under the heading "Postdocs with access to PDOs were more satisfied with their institution's response to the pandemic"– lines 251-281: "Postdocs with access to mental health resources were more likely to have their basic needs met"– Lines 284-298: "Postdocs were less confident in achieving their career goals than before the pandemic"This may require changing the order in which some of the results are discussed so that those reporting on a related topic are more clearly grouped together. For example, all the data about the benefit of PDOs/PDAs could be grouped into a single section, and the data on basic needs (lines 153-162 and 209-219) could also be grouped together.The above examples are not exhaustive and are just suggestions. Please use your own judgment to decide which findings to group together and the main result you would like to highlight in the relevant subheading.

We would like to thank the reviewer for these comments and giving us the opportunity to clarify the Results section. The Results section was edited, and additional headings were added:

“COVID-19 Impact on Postdoc Well-being” New line: 121-157“Institutional Responses to COVID-19 and Impact of PDAs and PDOs” New line: 158-175“Meeting Basic Needs During the Pandemic” New line: 176-239“Effect of Institutional Support on Mental Health and Wellness” New line: 255-295

For consistency, topic headings rather than conclusion statements were used for the headings.

2) The discussion frequently restates the results rather than interprets them (for example: Page 17, Lines 351 to 357; Page 18, Lines 375-378, 379-380; and Page 19, Lines 398-403, 406-407).Please reduce these repetitions and expand on the points listed below:

Repetition of the Results in the conclusion section was initially included to emphasize the key findings, but we acknowledge that this is redundant. In order to reduce repetition and improve the focus and flow of the Discussion, we deleted:

Lines 354-357 (original/submitted version)

“Unsurprisingly, given that the pandemic survey was conducted in a subset of the pre-pandemic survey, the demographics were comparable between the two surveys, with the exception of the respondents being older and further along in their careers, as expected.”

Lines 379-380 (original/submitted version)

“As previously indicated, this survey provides a unique “before-and-during” opportunity to observe the effects of COVID-19 on postdoctoral life.”

Lines 401-403 (original/submitted version)

“Here we’ve shown that nearly a quarter of all postdocs felt that their mental health needs were unmet during the pandemic and just as unsettling, a non-negligible proportion struggled with access to food (2%) and healthcare (7%).”

Lines 406-407 (original/submitted version)

“Moreover, respondents that were no longer in postdoctoral positions due to the pandemic had higher rates of unemployment. We did not collect detailed information about these former postdocs and more follow-up studies are needed to track their outcomes.”

We also moved lines 375-378 (original/submitted version) to the Results section (New lines 127-130).

“2,768 comments were collected in total, which further demonstrates the impact of the pandemic on the postdoctoral population.”

i) As pointed out in line 364, there have now been a few reports about the pandemic's effect on the workforce, including postdocs. How does the data fit with this literature, and altogether what is the takeaway message with respect to the new findings presented here? How do you interpret the demographic differences observed?

To acknowledge several new reports on the impact of the pandemic on the STEM workforce, additional references have been added to the Discussion. Our findings highlight the importance of institutional resources for the mental health and well-being of the postdoctoral population. This is our main takeaway message – “As we^22^ and others have previously reported, postdocs are an often overlooked and forgotten population in academia, with a non-negligible number of institutions being unaware of their total postdoc population, let alone the concerns of postdocs. Access to employee benefits varies widely^1-3^, with untoward consequences for postdocs who lack access to certain essential services. In a position that emphasizes sacrifice for research, institutions should be more proactive in discerning the basic needs of their postdocs, and make appropriate resources and benefits available, including access to paid family leave, healthcare, mental health and wellness services for the postdocs and their families, as well as affordable childcare options. Offices of Postdoctoral Affairs are the appropriate institutional entities to ensure that postdocs’ minimal basic needs are met and should assume responsibility for coordinating and disseminating these essential benefits.” (New lines 445-455) and we have highlighted this further in the Discussion.

(New lines 377-382)

“As is apparent from our survey, access to institutional resources like PDAs and PDOs is important not only for the ability of postdocs to complete their work in safe and supportive environments, as is often the focus of institutional efforts, but also for their mental and physical wellbeing. This intriguing correlation between postdocs’ satisfaction with their institution's response to the pandemic, and specifically whether their mental health needs were met, are important aspects that need further examination.”

We found few demographic differences especially with abilities to meet different basic and mental health needs during the pandemic. While these point towards an alarming trend, it would be premature for us to interpret any demographic differences observed. We hope that our work will encourage future studies to explore the long-term effects of the pandemic on specific demographics.

ii) How the challenges faced by postdocs compare to the rest of academia.

This is an important discussion point and we appreciate the reviewers giving us the opportunity to expand beyond our focus on the burdens of postdocs. Additional references were added to the Introduction and Discussion in order to emphasize the challenges faced by postdocs and others in the academic environment.

New lines (22-25):

“The COVID-19 pandemic has only made this status worse for postdocs due to lab closures, rotating (work) schedules, hiring and salary freezes, etc.; and especially for postdocs with families since school and daycare closures significantly disrupted research continuity ^4,5^.”

New Lines (362-363):

“Although there have been multiple reports of the pandemic’s impact on the STEM workforce^13-16,24-27^ few have discussed postdocs specifically^21,27^.”

iii) In Line 409 - Can you be more specific about "hints of long-term consequences"? What are you suggesting would be the long-term consequences of reduced productivity, delayed job searches, etc?

This is a very important question, which deserves further attention, and new text and references were added to suggest potential long-term effects. Our intent is to follow up on these outcomes in future studies to directly address the outcomes as the pandemic subsides.

New Lines (386-396):

“These experiences, especially combined with the frequently reported increase in mental health concerns, may result in more postdocs becoming disenchanted with their career trajectories. The long-term effects of the pandemic on the postdoctoral population, however, are unclear from our data, which was only conducted over a short period of time during the pandemic and further studies are needed to assess how the pandemic ultimately affected career trajectories. Although it is difficult from current data and probably premature to predict irreversible trends: the recent study ^28^ on the previously discussed gender gap in academic publications during the pandemic ^29-40^ suggests that long-term outcomes may be more nuanced than previously reported. While we are barely out of the COVID-induced disruptions, some glimpses of positive changes are beginning to come forth ^41-43^.”

iv) Do your data speak to irreversible trends that will impact the workforce, or rather do they suggest areas where specific interventions could be impactful/warranted? What are the future directions of this work?

Please see point iii above, that also addresses irreversible trends. Suggestions for future studies, which we plan to pursue, have been added in several paragraphs in the expanded limitations section.

New lines (377-382):

“As is apparent from our survey, access to institutional resources like PDAs and PDOs is important not only for the ability of postdocs to complete their work in safe and supportive environments, as is often the focus of institutional efforts, but also for their mental and physical wellbeing. This intriguing correlation between postdocs’ satisfaction with their institution's response to the pandemic, specifically whether their mental health, childcare and basic needs were met, are important aspects that need further examination.”

New lines (402-406):

“Since PDOs often provide professional development programming, etc., these findings suggest a potential reason why postdocs nationally were significantly more satisfied in the presence of these institutional assets, which would have been responsible for enhanced remote programming; again, an important factor that deserves more exploration.”

New lines (433-435):

“We chose a period (Fall 2020) when the urgency and uncertainty of the initial shutdown period had lessened. However, this only provides a small snapshot in time during the pandemic.”

New lines (436-441):

“We also did not directly analyze how the experience varied for postdocs in different fields. Some postdocs were likely able to transition readily to remote work for a period of time, with little to no disruption to their research (e.g., Biostatisticians), while others may have had their research productivity significantly hindered or disrupted due to limited access to their laboratories. These are all important factors that will need to be evaluated in future studies.”

v) Were there any positive impacts of the pandemic?

Although we would not necessarily suggest that the pandemic had any positive impacts, there were some noted examples where the consequences of the pandemic resulted in positive changes. To address this question, we have added potential positive impacts and additional references for context to the Discussion.

New lines (442-444):

“Lastly, further studies should also ascertain whether the pandemic had any positive impacts on postdocs, such as the ability to continue to work remotely under certain circumstances or make new professional connections via video conferencing, etc.^8,41-43^.”

New lines (394-402):

“While we are barely out of the COVID-induced disruptions, some glimpses of positive changes are beginning to come forth ^41-43^. As an example, in order to mitigate isolation and perpetuate a learning environment when labs shut down in early 2020, institutional entities responsible for professional skill-building and career-promoting programming pivoted to remote platforms with some interesting outcomes^16^. For instance, remote sessions now appear to be not only acceptable, but may even be preferred by trainees, as they are more accessible, provide more efficient use of time and offer flexibility (such as having the possibility to watch recorded sessions later)^16^.”

3) Please add a Limitations section which should include the following points (as well as lines 379-393 in the Discussion):

We thank reviewers for noting that our previous limitations section was not clear enough, thus, we have expanded on the limitations section in the Discussion as requested. Please see details below.

i) The pandemic was experienced very differently and at different times among different geographic locations in the United States. There is no mention of this or how it may affect the conclusions of the manuscript since this is not a nationally representative sample of postdocs.

This reviewer’s comment is valid as the pandemic has had a wide range of consequences individually. Although we were underpowered to do analyses based on geographic location within the US, we did limit the window of response to just a 1-month period so that responses would be more reflective of a specific point in time in the pandemic. Additionally, all of these variables have now been included in the Discussion as requested.

New lines (424-436):

“Individual pandemic experiences varied greatly throughout the pandemic, undoubtedly influenced by factors such as geographic location, type of institution, timeframe within the pandemic, caregiving responsibilities, and type of research which was amenable to remote work. Our study did not assess whether there was an unequal distribution of the effects across the country based on geographic location or type of institution (for example, an institute with more resources, budget, and personnel dedicated to postdoc affairs vs. others) within the United States. Nevertheless, we did capture responses from postdocs across the US, with higher response rates in locations that we expect would have more postdocs, such as on the East Coast and in California (Figure 1 - supplementary figure 1A-B). We chose a period (Fall 2020) when the urgency and uncertainty of the initial shutdown period had lessened. However, this only provides a small snapshot in time during the pandemic. Unfortunately, there have been successive waves of the pandemic, the effects of which will require further exploration.”

ii) The impact of the pandemic was different for postdocs in different fields. For example, basic scientists who had their labs shut down were in very different situations than biostatisticians who could more easily continue to work from home

We acknowledge this important variable which is now mentioned in the discussion.

New lines (436-441):

“We also did not directly analyze how the experience varied for postdocs in different fields. Some postdocs were likely able to transition readily to remote work for a period of time, with little to no disruption to their research (e.g., Biostatisticians), while others may have had their research productivity significantly hindered or disrupted due to limited access to their laboratories. These are all important factors that will need to be evaluated in future studies.”

Furthermore, as mentioned above, we acknowledge that individual, institutional and regional differences in COVID response have made the pandemic a very different experience for everyone involved. It is our hope that by including quotes of individual responses to open-ended questions in the survey, we address these wide ranges of experiences.

iii) Potential for response bias. Because the pandemic survey was only sent to those who provided email addresses, it raises concerns that the people who responded were inherently different than those who did not provide an email address. Additionally, people who were unemployed may have been more likely to spend time answering the pandemic survey than those who were employed

As with any survey-based research, response bias is a concern, and it is possible that certain aspects of these results may be skewed. However, given that the second survey was conducted in a subset of those individuals who responded to the initial survey and analyses were primarily conducted as comparisons between the two surveys, we anticipate that potential bias will be mitigated in our study design. Furthermore, to clarify, 5,594 email addresses were collected in 2019, representing over 89% of all survey respondents, thus the pandemic survey was distributed to the majority of pre-pandemic survey respondents. This is now clarified in the Methods section (lines 454 and 466 in the original/submitted version). Many emails “bounced” -likely due to postdocs having left their institution by the time the pandemic survey was issued–and we were unable to follow up with these respondents. Those respondents who were no longer postdocs, whether they were newly employed in positions other than as a postdoc or unemployed, were analyzed separately in a different section of the results. This has been clarified in the Results and Methods sections.

New lines (498-500):

“All responses were collected anonymously, but most respondents (5,594 (89%)) voluntarily provided contact information…”

New lines (509-516):

“All postdocs who completed the initial pre-pandemic survey and submitted their email addresses for recontact (5,594 total postdocs) were asked to complete this second pandemic survey, which was launched on October 1, 2020 and was only open for one month. Many emails bounced, suggesting that many postdocs from the previous cohort had defunct addresses. In total, 1,942 responses to the pandemic survey were collected. Of these responses, 1,722 (88.6%) were submitted by researchers currently in postdoctoral positions in the United States, and these responses are analyzed here.”

iv) A main take home that the authors point out is that the presence of Postdoc Offices or Postdoc Associations as a positive impactor on some challenges faced by responders. However, based on the current study this claim is only correlative. For example, it is unclear if the respondents had accessed their local PDO/PDA or had just heard of them. Furthermore, various other contributors are also not included in the present study, such as type of institution (availability or abundance of resources, personnel dedicated to postdoc affairs, budgets, etc.), geographical location of the institution, number of postdocs within the institute, location of PDO (university-wide or college-specific), disciplinary area of research, and community and camaraderie among colleagues.

We agree with the reviewer that our findings are correlative, as are most survey-based research findings. We have addressed the text so that this is clear throughout the paper.

This sentence has been added to the discussion and the caveats are further discussed in the section that follows.

New lines (424-427):

“Individual pandemic experiences varied greatly throughout the pandemic, undoubtedly influenced by factors such as geographic location, type of institution, timeframe within the pandemic, caregiving responsibilities, and type of research which was amenable to remote work.”

As suggested by the reviewers, we agree that many other factors could have impacted how each postdoc experienced the pandemic and the presence of PDA/PDO is not the only positive impactor. It is also true that the presence of a PDA/PDO itself reflects the availability or abundance of resources, personnel, and budgets at an institute. Investigating these individual factors is beyond the scope of our present study and we have emphasized in the Discussion that future studies should address this issue. While we cannot conclude if our respondents had only heard of the resource or had actually used their resources, our correlative findings indicate that the presence of PDA/PDOs at the institute has had a positive impact during the pandemic. It highlights their importance, the work they do and resources they provide to enhance postdoc wellness, and institutions that do not already do so should dedicate budgets and personnel for these assets.

4) The article concludes that the pandemic profoundly impacted the career trajectories of postdocs. However, the long-term impact of the pandemic on the career trajectories of postdocs is not clear from the current data, since the survey results only represent a single snapshot in time during the first year of the pandemic.Please can you tone down this statement and include the caveat that the current data only represents a snapshot in time. Furthermore, are there any other studies that support this claim?

There have not been any studies that we are aware of that have reported on how the pandemic has affected the career trajectories of postdocs, although there have been some reports on its consequences on faculty, which we have referenced. We have tempered this statement by adding the following sentence.

New lines (388-391)

“The long-term effects of the pandemic on the postdoctoral population, however, are unclear from our data, which was only conducted over a short period of time during the pandemic and further studies are needed to assess how the pandemic ultimately affected career trajectories.”

5) Please update the definition for underrepresented that is used in this manuscript and analyses (pages 22-23) so it is consistent with the NIH definition: https://diversity.nih.gov/about-us/population-underrepresented.

The NIH definition for racial categories was used and is now cited in the Methods section as requested.

New lines (520-531)

“Participants were queried about their race and ethnicity with the following options: white/Caucasian, Asian/Asian American, South Asian/South East Asian, Black/African American, Hispanic/Latino, Middle Eastern, Native American or Alaska Native, and Pacific Islander or Hawaiian Native. For all analyses, due to low numbers of respondents, certain racial/ethnic categories were consolidated into three groups: (1) underrepresented minorities (URM) as defined by the NIH (https://diversity.nih.gov/about-us/population-underrepresented): Black or African American, Hispanic or Latino, Native American or Alaska Native, Native Hawaiian or Pacific Islander; (2), Asians (Asian/Asian American and South Asian/Southeast Asian); and (3) white (white/Caucasian and Middle Eastern). The NIH also defines those with disabilities (analyzed separately here) and those from disadvantaged backgrounds (this category was not queried in this study) to also be underrepresented in the biomedical, clinical, behavioral, and social sciences.”

6) The paper contains a number of univariate analyses. Please can you indicate whether or not you adjusted your p-values for multiple comparisons?

Based on comment number 8 below, we reanalyzed our data using multivariate analysis, including multiple variables in the model: Gender, Race and Ethnicity, Residency, LGBTQ status, having a disability, access to a PDO and access to a PDA. This allowed us to observe new results (see the response to comment #8) and avoid adjusting for multiple comparisons.

7) Figure 1 - figure supplement 1: In C, D, E the bar heights don't match the number labels (%)- e.g. all bars that say 1% should be the same height, bars that say 2% should be the same height, bars that say 0% should be the same height, bars that say 4% should be the same height. The bar marked 8% in panel D doesn't match where 8% would be on the y axis, etc. Please double-check the data and replot or relabel as appropriate.

We would like to thank the reviewer for identifying these discrepancies. More precise percentages were added in the figures to better match the y-axis. Please see updated Figure 1-figure supplement 1 C, D, E, F, G. Other supplementary figures were also modified for consistency (Figure 2-figure supplement 1, Figure 3-figure supplement 1, Figure 4-figure supplement 1 and Figure 5-figure supplement 1).

8) Your analyses could be strengthened by using multivariate ordinal logistic regression models and/or logistic regression models. On page 9, line 168, you state that the majority of Asian post-docs were international and more likely to have unmet healthcare needs. If you created a multivariate logistic regression model, the reader would be able to determine whether race or international status (and other factors) were independently associated with unmet needs. Please consider doing this for some or all of the analyses; however, we do not insist on this.

We would like to thank the reviewers for these suggestions. We re-analyzed our data using either a multivariate logistic regression model or a multivariate ordinal logistic regression model, when appropriate. The majority of the results were strengthened, and we also observed newly significant ones. The new results are now specified in the Figure legends for Figure 2F; Figure 2- Figure supplement 1 A; Figure 3A; Figure 3- Figure supplement 1 A, B, C, D; Figure 4

New significant results:

Figure 2 F, having access to a PDA significantly impacted having mental health needs met.Figure 4E-F, a smaller portion of postdocs unaware of mental health resources had their mental health needs or all their basic needs met.

We also added new supplementary Figures due to new findings:

Figure 2 Figure supplement 1 B. Using the multivariate logistic regression model, we observed significantly unmet basic needs based on LGBTQ status.Figure 2 Figure supplement 1 C. Using the multivariate logistic regression model, we observed significantly unmet basic needs based on disability status.Figure 3 Figure supplement 1 C and D. Using the multivariate logistic regression model, we observed significant differences in Immigration concerns according to race and ethnicity (C) and LGBTQ status (D).Figure 5 - Supplementary 1C-D: Significant results according to geographic constraints

Some results did not remain significant, mainly due to confounding and were removed from the paper.

Figure 2 D, postdoc unaware of PDO did not significantly differ from postdoc with a PDO regarding satisfaction of an institutional response.Figure 2F, having access to a PDA did not impact having all basic needs met when using the multivariate analysis.Figure 3A, no more significant differences between citizenship status and health care needs met. The effect was driven by Asian postdocs.Figure 4B, no difference observed between non-binary/third gender and male for the impact on mental health and use of institutional resources.Figure 4B, no significant differences were observed according to residency.Figure 4B, Postdocs with disabilities did not use more mental health resources.Figure 4H, Postdocs having access to a PDA and using mental health resources is not significant anymore.Figure 4- Supplementary 1, Postdocs unaware of mental health resources did not differ significantly from those having access according to food and health care needs met. The postdocs with no access to mental health resources remains significantly different from the ones that have access to these resources.Figure 5 - Supplementary 1C, no more difference according to peer pressure, was confounded with residency statusFigure 5 - Supplementary 1E, no more difference in job market perception according to race/ethnicity. The figure was removed.

We also added the models in the method section.

New lines (534-544)

“To assess similarities between the pre-pandemic and pandemic surveys, we used the Chi-square test in the presence of categorical data (basic R function). We used the same test to assess the postdocs who transitioned out of their position during the pandemic, due to the lack of additional data available for this group. For the other analyses, we performed multivariate tests including gender, race and ethnicity, residency, LGBTQ status, having a disability, access to a PDO and access to a PDA as covariates. To assess differences, we used either ordinal logistic regression in the presence of ordinal dependent variables (using the R package “MASS”), logistic regression (basic R function) and multinomial logistic regression (using R package “nnet”) accordingly. Results are described in the legend as Odds Ratio (OR) [95% Confidence Interval (CI)] with the p-value.”

9) Fig 5D/Supplemental Fig 5: it would be interesting to see what these data look like by gender, particularly as it appears to be a general national trend that women are leaving the workforce at higher rates and experiencing particular challenges balancing family and career. Again, please consider doing this analysis; however, we do not insist on this.

Fig 5D showed that a large % of postdocs changed their career plans (34%) of which the majority was due to the pandemic. However, these changes in plans did not differ by gender.

These results were mentioned in our final sentence of the paragraph.

New lines (327-329):

“Lastly, no differences were observed by gender or identity groups with respect to reasons for changing career trajectory (Multivariate logistic regression, p>0.05, data not shown).”

Additionally, no differences by gender were observed for reasons for changing career trajectory (New lines 333-334), or job market perception change (New lines 327-329).